# Vaccination Coverage among Immunocompromised Patients in a Large Health Maintenance Organization: Findings from a Novel Computerized Registry

**DOI:** 10.3390/vaccines10101654

**Published:** 2022-10-02

**Authors:** Shirley Shapiro Ben David, Iris Goren, Vered Mourad, Amos Cahan

**Affiliations:** 1Health Division, Maccabi Healthcare Services, Tel Aviv 6812509, Israel; 2Sackler Faculty of Medicine, Tel Aviv University, Tel Aviv 6997801, Israel; 3Infectious Diseases Unit, Samson Assuta Ashdod University Hospital, Ashdod 7747629, Israel

**Keywords:** immunocompromised, registry, vaccine, pneumococcal, influenza, meningococcal, hepatitis B, PCV13, PPV23

## Abstract

Immune-compromised patients (IPs) are at high risk for infections, some of which are preventable by vaccines. Specific vaccines are recommended for IP; however, the vaccination rate is suboptimal. The aim of this study is to describe the development of an IP registry and to assess vaccination rates in this population. A population-based registry of IPs was developed using an automated extraction of patient electronic health-record data in Maccabi Healthcare Services (MHS), an Israeli health maintenance organization serving over 2.4 million members. Included in the registry were patients receiving immunosuppressive therapy (IT); patients living with HIV (PLWH); solid organ and bone marrow transplant recipients (TR); patients with advanced kidney disease (AKD), and asplenic patients. We evaluated the full schedule for each vaccine’s uptake rates for influenza, pneumococcal, meningococcal, and hepatitis B. On 1 October 2019, 32,637 adult immune-compromised patients were identified by the registry. Of them, 1647 were PLWH; 2354 were asplenic; 5317 had AKD; 23,216 were on IT; and 1824 were TR. Their mean age was 57 and 52.4% were females. The crude rate of immune compromise among adult MHS members was 2%. Vaccine coverage rate was overall low for PCV13, with only 11.9% of all IPs in the registry having received one dose. Influenza and PPV23 vaccination rates were higher (45% and 39.4%, respectively). Only 5.3% of all IPs received all three vaccines. Overall, low vaccination coverage was found among IPs. Our registry can serve to identify target-patient populations for interventions and monitor their effectiveness.

## 1. Introduction

Immunosuppression can be the result of an underlying procedure, a disease, or its treatment. With more and more new and efficient therapies being introduced, many of which have immunosuppressive effects, immune-compromised patients are a constantly growing population [1]. This population has an increased risk for serious infectious diseases due to the diminished functioning of their immune systems [2]. Therefore, the optimal protection of this vulnerable group is of the utmost importance. Influenza and invasive pneumococcal diseases both have higher frequency and mortality rates in this population [3,4,5]; however, mortality may be prevented by vaccination.

Israeli National Guidelines include specific recommendations for the vaccination of immune-compromised patients, including for annual influenza, both pneumococcal conjugate (PCV13) and pneumococcal polysaccharide vaccines (PPV23), and meningococcal and hepatitis B vaccines, in line with international guidelines [6,7]. These vaccines are available free of charge, or at a substantial discount, in community clinics nationwide.

There are little data regarding vaccination rates among immune-compromised patients. Available evidence suggests that vaccine coverage rates are low, even in countries with well-functioning healthcare systems [8,9,10]. Prior studies are limited by small sample sizes and non-electronic data collection, which is often reliant on patients self-reporting vaccinations [11,12,13]. Unawareness in primary care providers may also be contributing to low vaccine uptake rates [14].

A medical registry is a collection of patients that qualify for a set of pre-specified conditions. Developing and maintaining a medical registry requires substantial effort [15]. In this study, we describe the development of an immune-compromised patients registry using an automated collection of individual-level information on relevant characteristics, such as diagnoses, procedures, and the use of medications, with the goal of including patients for whom special vaccination recommendations apply [16]. We also report on vaccination rates in this population.

## 2. Materials and Methods

### 2.1. Setting

Operating under the Israeli National Insurance act, Maccabi Healthcare Services (MHS) serves a population of 2.4 million members (about a quarter of the Israeli population) nationwide, with each member having a unique National identifying number. Electronic health records (EHR) of all members are stored in a nationwide centralized database and include comprehensive information (including demographics, diagnoses, immunizations, encounters, procedures, hospitalizations, and drug prescriptions and fillings, as well as laboratory tests).

The immune-compromised patients registry was developed using EHR data in order to identify in an automated manner (i.e., without the need for active reporting by physicians) patients with selected types of immune compromise.

A designated team developed the registry. Direct communication with the attending physician to validate the registry and improve patient follow-up was frequently performed. Connected to MHS clinical systems, the registry is automatically updated on a daily basis. Information about inclusion in the registry is displayed in the EHR as well as on the personal patient’s web health record.

### 2.2. Registry Population

Eligible for the registry were all MHS members aged 18 years or above meeting any of the following inclusion criteria:(1)Receiving immunosuppressive treatment in the previous six months, including chemotherapy and radiation therapy for oncologic patients;(2)Patients living with HIV (PLWH)—HIV diagnosis (according to the International Classification of Diseases version 9 with clinical modifications (ICD-9-CM) codes: 044.9, 043.9, 795.71, 043.2) with a record of central MHS preauthorization for HIV-specific combination antiretroviral treatment (patients who purchase antiretroviral treatment for pre- or post- HIV exposure prophylaxis were excluded);(3)Patients with advanced chronic kidney disease (CKD) identified based on two consecutive serum creatinine levels corresponding to an estimated glomerular filtration rate < 30 mL/min, or receiving renal replacement therapy for at least three consecutive months during the previous seven months;(4)Transplant recipients (TR)—patients who had received solid organ (kidney, liver, intestines, heart, lung, pancreas) and/or hematopoietic stem cell transplant (HSCT);(5)Asplenia—record of a splenectomy procedure or a physician diagnosis of asplenia (ICD-9-CM codes: 759.0, 41.5).

Registry entry date was the date of the first qualifying medical event.

### 2.3. Recommended Vaccines

We evaluated vaccine uptake rates for the following vaccines, recommended according to the national Israeli ministry of health guidelines: annual influenza vaccine; pneumococcal vaccine—one dose of PCV13, one dose of PPV23 plus booster under 65 years of age and/or one dose after 65 years of age; meningococcal conjugate vaccine (for patients with asplenia and patients living with HIV)—two doses of the conjugated vaccine for serogroups A, C, W, and Y; MenACWY (the meningococcal group B vaccine was not available in MHS during the development of the registry); and hepatitis B—at least 3 doses for patients living with HIV and patients on dialysis.

Annual influenza and PPV23 vaccines were available without a prescription or referral from a physician, and were free of charge for patients >65 years. PCV13, MenACWY, and hepatitis B vaccine administration required a prescription and were available for purchase in a pharmacy at a subsidized price.

### 2.4. Data Analysis

We extracted demographics, smoking status, body mass index (BMI), and vaccination status from her data. We defined presence of comorbidities based on criteria used in other MHS registries for chronic diseases (cardiovascular disease, hypertension, diabetes, chronic obstructive pulmonary disease, osteoporosis, inflammatory bowel disease, cognitive impairment). We based residential socioeconomic status on a score ranked from 1 (lowest) to 10, which was derived by the Israel Central Bureau of Statistics [17].

We anonymized data and analyzed using the Statistical Package for the Social Sciences (SPSS) version 25 (IBM^®^ SPSS^®^ Statistics). We used Student’s *t*-test to assess between-group differences. 

The study was approved by the local Institutional Review Board (IRB), approval number: 0015-20MHS. Informed consent was waived by the IRB, as all identifying details of the participants were removed before the computational analysis.

## 3. Results

On 1 October 2019, 32,637 adult immunocompromised patients were identified (Figure 1). Of them, 1647 were patients living with HIV, 2354 were asplenic, 5462 had advanced chronic kidney disease, and 1824 were transplant recipients. Additionally included in the registry were 23,216 patients on immunosuppressive treatment, and of them, 6647 were oncologic patients who received chemotherapy and/or radiation therapy. The crude rate of immune compromise among adult MHS members was 2%. Table 1 shows patient demographics.

Overall, 52.4% were females, with a lower proportion of women among patients living with HIV and transplant recipients (25.6% and 27.3%, respectively). The mean age of the patients included in the registry was 57 years, with a lower mean age among patients living with HIV (46.1 years) and higher among patients with advanced chronic kidney disease (73.4 years). The mean (±SD) length of time from the event qualifying for inclusion in the registry was 5.6 (±5.4) years.

The vaccine coverage rate was overall low for PCV13, with only 11.9% of all immune-compromised patients in the registry having received one dose (Figure 2). The vaccination rates for PPV23 as well as annual influenza (2018–2019) vaccine were higher, at 39.4% and 45%, respectively. Only 5.3% of all immunocompromised patients received all three vaccines as recommended by national guidelines (Table 2). Higher vaccination coverage rates were observed for PPV23 and also for annual influenza vaccine among patients aged ≥ 65 years than younger patients (79% vs. 16% (*p* < 0.001) and 65.4% vs. 34.3% (*p* < 0.001), respectively) but lower for PCV13 (8.5% vs. 13.9%; *p* < 0.001) (Table 2).

Patients living with HIV had a higher PCV13 uptake rate than the total registry population (34% vs. 11.9%; *p* < 0.001) but a lower uptake rate for PPV23 (32% vs. 39.4%; *p* < 0.001). In addition, a series of two doses of MenACWY vaccine was received by only 9.8% of this population, while three or more doses of hepatitis B vaccines were received by 27.1%.

Vaccination rates among asplenic patients were higher compared to the total registry population for PCV13, PPV23, and annual influenza (21.8%, 52.3%, and 54.8, respectively; *p* < 0.001). However, a series of two doses of MenACWY vaccine uptake rate was only 13.3%.

The lowest PCV13 uptake rate (6.8%) was observed among patients with advanced chronic kidney disease; however, this population had the highest PPV23 and annual influenza vaccination rates (67.5% and 63.6%, respectively). Among patients receiving renal replacement therapy, 39.6% had a record of completing at least three doses of the hepatitis B vaccine.

## 4. Discussion

The present study describes the development and validation of a novel registry of populations of immune-compromised patients in a large HMO in Israel. As of October 2019, the registry included 32,637 patients in five main categories. The point prevalence of immune compromise among adult MHS members was 2%. Vaccination rates were low for PCV13 (11.9%) and suboptimal for PCV23 (39.4%) as well as for the annual influenza vaccine (45.8%). The majority of immune-compromised patients did not receive any of the three assessed vaccines, with only about 5% fully vaccinated.

The poor adherence to guideline-recommended vaccines that we found in this population is concerning. These findings are in line with other studies on PLWH [18], chronic kidney disease and dialysis [19,20], TR [21], and patients with asplenia [22]. Particularly, patients on IT were found to be under-vaccinated compared with other IP populations, including patients on biological agents [23,24]. Other small studies showed low influenza and pneumococcal vaccination rates in cancer patients [25,26]. As in other populations, low vaccination rates among patients on IT may be explained by immunization hesitancy [27]. Our study also included patients with cancer. In these patients, uncertainty about the right timing to vaccinate relative to antineoplastic treatment may contribute to low vaccination rates.

In agreement with other studies [20,28], this study found higher influenza and PPV23 vaccination rates in the elderly. Age, being another indication for these vaccines, may be the reason for that higher coverage rate and not the immunosuppression status. In Israel, PPV23 vaccination in the elderly (including immunocompetent patients) is part of the national program for quality care in the community [29]. Indeed, in MHS there is an electronic alert reminding clinicians to consider PPV23 and influenza vaccinations in these patients, which may also contribute to the higher vaccination rates.

Clinician unawareness of a patient’s immune state may result from difficulty in accessing relevant information in the EHR and may contribute to incomplete vaccination. As immune compromise has many causes, constructing the registry required a careful examination and assessment of the variety of different conditions, procedures, and treatments leading to compromised immunity [30]. Updated regularly, the registry can help clinicians identify patients as having immune compromise and act accordingly [31,32]. A computerized registry can also support interventions, such as alerts suggesting the need for vaccination. Therefore, an immune-compromised patient registry may provide a platform for identifying patients in need of vaccination and for interventions to improve vaccination rates.

The study has some limitations. The immune-compromised patient registry is based on automated data extraction rather than reporting by clinicians. As such, it is less dependent on active clinician cooperation; however, some patients not fitting the inclusion criteria may have been misclassified. Information on vaccines received outside of the HMO was only partly available. However, the vast majority of vaccines are administered in primary care clinics, which are a part of the HMO, and transition between HMOs is uncommon. Lastly, differences between healthcare systems may affect the generalizability of our findings to other countries.

## 5. Conclusions

Overall, and across vaccines, low vaccine uptake rates among immunocompromised patients were found. Considerable improvements in vaccination rates are required to reduce the burden of vaccine-preventable diseases. Vaccination strategies for vulnerable populations are needed, and this up-to-date registry can serve as a valuable resource for targeted vaccination programs.

## Figures and Tables

**Figure 1 vaccines-10-01654-f001:**
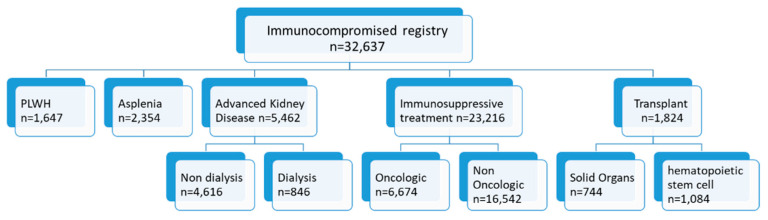
Immune-compromised patients registry structure and population, Israel, October 2019.

**Figure 2 vaccines-10-01654-f002:**
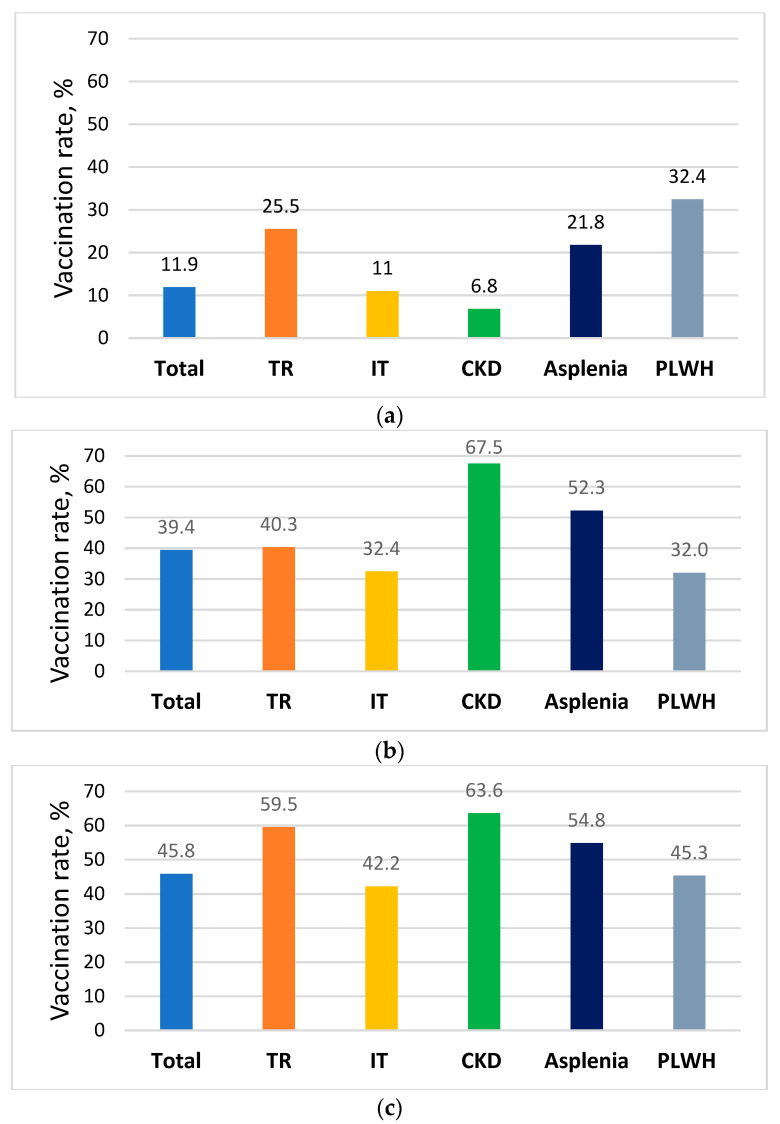
Pneumococcal and influenza vaccination rate uptake among immune-compromised patients in the immunocompromised registry, Israel, October 2019. (**a**) PCV13—pneumococcal conjugate vaccine; (**b**) PPV23—pneumococcal polysaccharide vaccine; (**c**) influenza seasonal vaccine, 2018–2019. Total immunocompromised patient registry, N = 32,637; TR—transplant recipients n = 1824; IT—immunosuppressive therapy, n = 23,216; CKD—advanced chronic kidney disease (GFR < 30 mL/min), n = 5462; asplenia n = 2354; PLWH—patients living with HIV, n = 1647.

**Table 1 vaccines-10-01654-t001:** Baseline characteristics of immune-compromised patient in the registry, October 2019.

Variable	Total n = 32,367	Transplant n = 1824	IT n = 22,851	Advance CKD * n = 5462	Asplenia n = 2354	PLWH n = 1647
Age (years)—mean (SD)	57 (17.4)	55.86 (13.3)	54.2 (16.6)	73.4 (13.9)	56 (15.1)	46.1 (10.6)
18–64 years no. (%)	20,502 (62.8)	1244 (68.3)	15,835 (69.3)	1180 (21.6)	1602 (68)	1560 (95)
≥ 65 years no. (%)	12,135 (37.2)	580 (31.7)	7016 (30.7)	4282 (78.4)	752 (32)	87 (5)
Gender, Female—no. (%)	17,087 (52.4)	751 (27.3)	13,113 (56.5)	2483 (45.5)	1122 (47.7)	421 (25.6)
Socioeconomic status ^‡^—no. (%)						
low	6146 (18.8)	332 (18.2)	3934 (16.9)	1335 (23.9)	455 (19.3)	462 (28.1)
med	15,938 (48.8)	846 (46.2)	11,124 (47.9)	2779 (50.9)	1136 (48.3)	857 (52)
high	10,552 (32.3)	650 (35.6)	8158 (35.1)	1250 (22.9)	773 (32.8)	328 (19.9)
Current Smoker—no. (%)	4226 (12.9)	192 (10.5)	2765 (12.1)	496 (9.1)	373 (15.8)	542 (32.9)
BMI—mean, SD	27.1 (5.8)	27 (5.2)	26.9 (5.7)	28.8 (6.2)	26.8 (5.9)	24.9 (5.1)
Comorbidities—no. (%)						
Obesity ^^^	8811 (27)	457 (25)	5841 (25.6)	2073 (38)	594 (25.2)	218 (13.2)
Underweight ^^^^	2264 (6.9)	110 (6)	193 (0.8)	27 (0.5)	153 (6.5)	172 (10.4)
Hypertension	13,036 (39.9)	917 (50.1)	7633 (33.4)	4436 (81.2)	800 (34)	211 (12.8)
Diabetes	6805 (20.9)	459 (25.1)	3624 (81.2)	2657 (48.6)	478 (20.3)	83 (5)
Cardiovascular disease	4532 (13.9)	290 (15.9)	2113 (9.2)	2117 (38.3)	280 (11.9)	64 (3.9)
COPD	1609 (4.9)	93 (4.6)	962 (4.2)	510 (9.3)	99 (4.2)	27 (1.6)
Cognitive impairment	722 (2.2)	19 (1)	292 (1.3)	383 (7)	42 (1.8)	5 (0.3)
Osteoporosis	6656 (20.4)	434 (23.7)	4609 (20.2)	1515 (27.7)	399 (16.9)	96 (5.8)
IBD	4384 (13.4)	34 (1.8)	4269 (18.7)	73 (1.3)	36 (1.5)	20 (1.2)
Time from entry to registry, (Yr)—mean (SD)	5.6 (5.4)	7.3 (5.4)	5.3 (5.4)	4.5 (4.6)	9.5 (5.7)	7.2 (4.4)

IT—immunosuppressive treatment; PLWH—patients living with HIV; CKD—advanced chronic kidney disease; Yr—years; SD—standard deviation; BMI—body mass index; CVD—cardiovascular disease; COPD—chronic obstructive pulmonary disease; IBD—inflammatory bowel disease. * CKD—estimated glomerular filtration rate, eGFR < 30. ^‡^ SES—socio-economic status (defined by the Israel Central Bureau of Statistics [17] as low, 1–3; medium, 4–6; high, 7–10. ^^^ Obesity—BMI kg/cm^2^ ≥ 30. ^^^^ Underweight—BMI < 17.

**Table 2 vaccines-10-01654-t002:** Immunocompromised patients in registry and vaccination rate, Israel, 2019.

	PCV13	PPV23	Influenza 2018–2019	Pneumo. + Annual Influenza	MenACWY *	Hepatitis B **
Total registry N = 32,637	3882 (11.9)	12,857 (39.4)	14,963 (45.8)	1728 (5.3)		
PLWH n = 1647	534 (32.4)	527 (32)	746 (45.3)		161 (9.8)	447 (27.1)
Asplenia n = 2354	513 (21.8)	1230 (52.3)	1291 (54.8)		314 (13.3)	
CKD n = 4616	201 (4.4)	3260 (70.6)	2916 (63.2)			
Dialysis n = 846	173 (20.4)	429 (50.7)	611 (70.7)			342 (40.4)
Non Oncologic IT n = 16,542	2113 (12.8)	4794 (29)	8648 (52.3)			
Oncologic IT n = 6674	498 (7.5)	2613 (39.2)	3641 (54.6)			
HSC TR n = 1084	224 (20.7)	422 (38.9)	612 (56.5)			
Solid organ TR n = 744	242 (32.5)	313 (42)	476 (48.3)			
18–64 years n = 20,502	2846 (13.9)	3275 (16)	7023 (34.3)			
65 years and Above n = 12,135	1036 (8.5)	9582 (79)	7940 (65.4)			

PCV13—pneumococcal conjugate vaccine; PPV23—pneumococcal polysaccharide vaccine; CKD—advanced chronic kidney disease (GFR < 30 mL/min); PLWH—patients living with HIV; IT—immunosuppressive therapy; HSC—hematopoietic stem cell; TR—transplant recipients. * Two doses of MenACWY and meningococcal ACWY conjugated vaccine were assessed only for the recommended targeted population. ** At least three doses of hepatitis B vaccine were assessed only for the recommended targeted population.

## Data Availability

Not applicable.

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
