# Peer review of "Vaccination Coverage among Immunocompromised Patients in a Large Health Maintenance Organization: Findings from a Novel Computerized Registry"

_vaccines, 2022, doi:10.3390/vaccines10101654_

Round 1
Reviewer 1 Report
This report presents vaccination coverage rates in immunocompromised patients based on a novel computerized registry in Israel. And the authors reported the low vaccination rates.
It is essential to prevent specific infectious diseases by vaccination, and the low rates are concerning.
The recommendation of vaccination for immunocompromised patients may differ within countries. In addition, those recommendations for immunocompromised patients are not easy to understand, different from the case of universal vaccines or vaccines for elderly populations.
Detecting immunocompromised patients is not easy for primary physicians. Therefore, the automated detection and alerts system would be beneficial.
As the authors stated in lines 225-226, misclassifications are essential critical factors when the automated alerts system is available. Therefore, whether these automated detections are correct between physicians' diagnoses must be evaluated in the future. Is there any plan for the evaluation of the automated classification system?
There is a duplicated paragraph in Discussion(Lines 223-238).
Reviewer 2 Report
1. General: since the IP registry and this study about its implementation was focussed primarily on vaccines that prevent respiratory and related diseases, this reviewer was surprised that Hib vaccination coverage among pediatric immunocompromised patients was not included.
2. Abstract: several issues need further clarification. The authors list 4 vaccines of interest, but then report coverage for only 2 of them, including 2 types of one of the vaccines, and do not clarify if coverage reported is for the full schedule for each vaccine. The authors list AKD patients as study subjects, then report the number of CKD patients. The acronym MHS is not defined.
3. Lines 15 and 60: different numbers of participants.
4. Lines 99-100: the conventional acronym for this vaccine is MenACWY.
5. Discussion: the authors should comment on the absence of coverage data for MenACWY and HepB vaccines in certain categories of immunocompromised patients, and on the low coverage rates in the categories that were assessed.
6. Lines 201-202: the authors should expand this issue – are the clinicians reluctant to vaccinate these patients or are the patients reluctant to receive vaccination? This is critical information for the development of strategies to raise coverage.
7. Lines 213-214: it is not clear why a clinician would have difficulty accessing the vaccination status of a patient now, since the registry has presumably been established for a few years.
8. Lines 223-229: the authors should be more specific regarding the type and estimated proportion of vaccine doses that may not be recorded in the registry, in addition to those received outside HMOs. Are doses received before the patient became immunocompromised included retrospectively in the registry? Do all clinicians record vaccinations on EHR records?
9. Conclusions: the authors should add recommendations to HMOs and the Ministry of Health on practical, cost-effective strategies to resolve the low vaccination coverage rates revealed in this study. Strategies may differ for different categories of patients, or for different vaccines. This should be linked to the issue raised in comment 6 above.
10. References: depending on the journal’s editorial policy, it is standard practice these days to include a weblink or DoI reference, where possible, to provide ease of access for all references.
Reviewer 3 Report
The manuscript entitled “Vaccination coverage among immunocompromised patients in a large health maintenance organization, findings from a novel computerized registry” sheds a piece of important information about the vaccination status of the immunocompromised patients in Israel state health-care setting with the utilization of patient electronic health record data. The study exposed that the vaccine uptake rate among the immunocompromised individuals was appalling, with only 5.3% of the population receiving all the recommended vaccines included in the study. Although the manuscript establishes important data for the scientific community, there are a few suggestions to improve this work as given below.
Minor Comments:
1. Line 66 to 68 can be rephrased to better convey the meaning.
2. Line 66 to 68 - Kindly elaborate on the “automated manner” and how it was done.
3. Kindly make sure that the Table legend should be distinct.
4. Rephrase line 207. “Indication” may not be the right word.
5. Line 231-238. Paragraph repeated.
6. The graphs in picture 2 can be aligned properly with the changes in the design to make them more appealing
Major Comments:
1. The sample size of 32 thousand participants seemed to be less to this reviewer, whereas similar works involved more than three times of the number of participants, around one lakh.
2. The discussion part is lacking a detailed overview of the possible reasons for the lack of vaccination in IP despite the Israeli National guidelines’ recommendations for vaccination.
3. The conclusion may include a better description of the relevance and need of a medical registry of immunocompromised patients, and future measurements of the low rates of vaccination in IP.
The information on vaccinations received outside of the health maintenance organization was only partially available for the study, which may be the constraint on achieving the study's objectives despite the majority of vaccines administered in primary care clinics which are a part of the HMO.
